# Application of computer vision in assessing crop abiotic stress: A systematic review

**Nabil Anan Orka** [1]*, **Fardeen Md. Toushique**[1], **M. Nazim Uddin**[2], **M. Latiful Bari** [3]

**1** Department of Electrical and Electronic Engineering, Islamic University of Technology (IUT), Gazipur, Bangladesh, **2** Horticultural Research Centre (HRC), Bangladesh Agricultural Research Institute (BARI), Gazipur, Bangladesh, **3** Food, Nutrition, and Agriculture Research Laboratory, Centre for Advanced Research in Sciences, University of Dhaka (DU), Dhaka, Bangladesh

* nabilanan@iut-dhaka.edu

## Abstract

### Background

Abiotic stressors impair crop yields and growth potential. Despite recent developments, no comprehensive literature review on crop abiotic stress assessment employing deep learning exists. Unlike conventional approaches, deep learning-based computer vision techniques can be employed in farming to offer a non-evasive and practical alternative.

### Methods

We conducted a systematic review using the revised Preferred Reporting Items for Systematic Reviews and Meta-Analyses (PRISMA) statement to assemble the articles on the specified topic. We confined our scope to deep learning-related journal articles that focused on classifying crop abiotic stresses. To understand the current state, we evaluated articles published in the preceding ten years, beginning in 2012 and ending on December 18, 2022.

### Results

After the screening, risk of bias, and certainty assessment using the PRISMA checklist, our systematic search yielded 14 publications. We presented the selected papers through in-depth discussion and analysis, highlighting current trends.

### Conclusion

Even though research on the domain is scarce, we encountered 11 abiotic stressors across 7 crops. Pre-trained networks dominate the field, yet many architectures remain unexplored. We found several research gaps that future efforts may fill.

## Introduction

Abiotic stressors refer to environmental variables that adversely affect the growth and development of plants and reduce yield to suboptimal levels [1]. The majority of agricultural

**Data Availability Statement:** All relevant data are within the paper and its Supporting information files.

**Funding:** The author(s) received no specific funding for this work.

**Competing interests:** The authors have declared that no competing interests exist.

fields and breeding nurseries expose their plants to a wide range of abiotic stimuli, ranging both in type and severity, such as anomalies in light, radiation, extreme temperature, drought, chemical substances, salinity, an absence of essential nutrients, harmful gases, as well as other common to everyday stressors [2]. Abiotic stressors have the greatest impact on crops or commercial plants since they can decrease crop production by up to 70% and lead many crops to function at only 30% of their genetic potential [3]. The ongoing decline of farmlands, the reduction of water supplies, and the expanding global warming patterns and climate change caused a rise in yield loss, culminating in catastrophic financial damage [1].

In most cases, experts handle the painstaking task of identifying and classifying crop stress in two ways: destructive (laboratory analysis) and non-destructive (naturalistic observation). This manual approach is time intensive and prone to errors due to the subjective aspect of each professional's expertise and judgment [4]. Contrarily, computer vision technologies provision non-contact and effective solutions in agriculture, notably in the management of weeds, animals, and plants [5]. Deep learning in computer vision has increased significantly over the past few years, merging computer science with several disciplines of physical and life sciences [6]. When it comes to using computers to understand images or videos, both machine learning and deep learning can be used. However, a caveat of traditional machine learning is that it requires domain specialists to manually extract features that the computer can understand and work with [7]. Deep learning, on the other hand, can discern complex patterns in high-dimensional data with minimal feature engineering [7]. Deep convolutional neural networks (DCNN), for example, learn to map relevant features without human interference [8].

Although deep learning has been widely employed in agriculture, there is no unified compilation of research that highlights the effectiveness of various approaches, especially in crop abiotic stress recognition. This research explores the implementation of computer vision in crop physiology and offers an in-depth review of all articles on abiotic stress classification using deep learning.

## Materials and methods

### Registration and protocol

For reporting the findings, we followed the revised Preferred Reporting Items for Systematic Reviews and Meta-Analyses (PRISMA) guidelines [9] (see S1 File for the checklist). The study protocol was not entered into any registry. The research question was: To what extent, and how effectively, have the most recent deep learning advances in computer vision been integrated into crop abiotic stress assessment?

### Eligibility criteria

We confined the search period from 2012 through 2022 to uncover recent advances in crop abiotic stress recognition using computer vision, implying a distribution of studies in the previous decade. The following constitutes the inclusion and exclusion criteria -

Inclusion criteria:

- Original research articles written in English and published in peer-reviewed journals.

- Studies incorporating deep learning algorithms to identify or evaluate various abiotic stressors of crops such as nutrient shortages, drought, chemical injuries, and pH, among others.

Exclusion criteria:

- Research on plants that cannot be classified as crops.

- Research that does not focus exclusively on visuals or pictures as data input.

## Information sources

As per Gusenbauer et al. [10], only 14 of the 28 academic search systems reviewed are well-suited to evidence synthesis in the form of systematic reviews, having satisfied all required performance standards. As our investigated topic is reliant on many different subjects such as plant science, and computer science, we selected the following multidisciplinary principal databases mentioned in the study -

- Scopus

- Web of Science

- ScienceDirect

- Bielefeld Academic Search Engine (BASE)

- Wiley Online Library

On December 18, 2022, using the optimal search string described in the next section, we retrieved the results from various search engines. It should be noted that Scopus was the primary search engine utilized to determine the most effective search term.

## Search strategy

Before commencing the systematic study, it is critical to select the best search string possible in order to generate the maximum number of relevant articles that can address the issue at hand. The optimal search string, determined after several trials and errors, is the following—*("crop" OR "plant" OR "soil organic matter" OR "pH" OR "water") AND ("nutri*" OR "npk" OR "element*" OR "abiotic stress") AND ("defici*" OR "short*" OR "inadeq*" OR "insuffi*") AND ("estima*" OR "predict*" OR "recogn*" OR "detect*" OR "assess*" OR "analysis") AND ((("machine" OR "deep") AND "learning") OR "artificial intelligence" OR "image process*")*. We included 'plant' in our search phrase since some studies specified plants as their researched item, but, those can be referred to as crops.

S2 File includes the exact search strings of each search engine, along with additional parameters. The file includes certain exceptions. For example, the optimized search query returned no items in the BASE, therefore we dropped certain phrases to gather valuable documents. Moreover, due to Boolean connector constraints, we were unable to use the whole search string in the ScienceDirect engine, forcing us to discard a few keywords. Finally, we defined Agriculture as the subject area for ScienceDirect, BASE, and Wiley Online Library, which allowed us to exempt irrelevant entries.

## Selection process

Following the search procedure, all returned entries from the five search engines were added to a single comma-separated value (CSV) file, and duplicates were deleted. Two researchers (N.A.O. and F.M.T.) independently assessed all non-duplicate records' titles and abstracts using the eligibility criteria. A third researcher (M.N.U.) served as a mediator in the event of a

**Table 1. List of data extraction form items.**

| Data item | Description |
|---|---|
| Reference | Title, author, and year |
| Aim | Investigated abiotic stressors and crops |
| Dataset | Collection environment, sample per class, the color space of the input image, and the availability of the dataset |
| Approach | Dataset split ratio, learning type, and the utilized deep learning model |
| Outcome | Inter-class precision and recall on the test/validation data |

disagreement. Following that, two researchers (N.A.O. and M.N.U.) collaborated to screen the previously selected records for the eventual inclusion in the review by perusing their full texts.

## Data collection process

We developed a data extraction form to collect data from selected papers, which was utilized by a review author (N.A.O.) initially and then validated by another author (M.N.U.). All disagreements were aired and resolved collectively. S3 File contains the data extraction form.

## Data items

Table 1 lists and defines the collected data items. In general, the dataset split ratio is clearly stated in the article, i.e., the ratio of the entire dataset divided into training, testing, and validation data. We determined the split ratio manually for the studies that did not provide the ratio, but rather the sample sizes of each data category.

We selected the model that the authors identified as the best performing from said research and verified it using the reported overall model accuracy. Following that, we calculated inter-class precision and recall for each abiotic stress using the confusion matrix presented in the study. Although accuracy represents the model's overall efficacy, in an imbalanced dataset where the sample distribution of classes is not uniform, it skews towards the class that has the highest number of samples, assigning larger weights to it [11]. Moreover, while some researchers trained their algorithms to identify both biotic and abiotic stressors, we primarily looked for the latter in this systematic review. Owing to these reasons, we opted to retrieve precision and recall (see Eqs (1) and (2)), which indicate the model's ability to predict and recognize a specific class, respectively.

$$\text{Precision} = \frac{\text{True Positives}}{\text{True Positives} + \text{False Positives}} \tag{1}$$

$$\text{Recall} = \frac{\text{True Positives}}{\text{True Positives} + \text{False Negatives}} \tag{2}$$

## Study risk of bias assessment

After reviewing the full texts to the eligibility requirements, the selected studies, a total of 26 publications, were subjected to a two-round risk of bias analysis. In the first round, we identified whether the publisher or journal is predatory using a decision tree (see S1 Fig) derived from several prior studies [12–16] and the checklist from the "*Think. Check. Submit*" campaign [17]. While conducting a systematic review, encountering eligible articles published in

potentially predatory journals is increasingly prevalent [18]. Since the utility of a systematic review is dependent on the articles evaluated, it is critical to exclude the predatory journals that lessen reliability [18]. Two reviewers (N.A.O. and M.N.U.) collaborated to complete the screening and eliminated the doubtful articles from further consideration in this review. The first round's five discarded articles are included in the S4 File, leaving 21 articles for the next round.

In the second round, we evaluated the methodological quality of the remaining 21 studies to minimize the chance of selecting studies with limited evidence. We used a slightly modified quality assessment tool adopted from [19, 20]. The scoring technique was revised to promote flexibility and inclusiveness, as binary grading on a criterion fulfillment appeared harsh in general. This framework focuses on the primary outcome, thoroughness of the literature review, validation of the studied dataset, reproducibility of the framework with simple explanations, and a coherent conclusion supported by the outcome (see Table 2). Overall methodological quality scores were determined by adding each study's specific criteria ratings. The methodological quality was rated as "high" if the overall score was greater than or equal to 4, "moderate" if the overall score was less than 4 but greater than 2, and "low" if the overall score was less than or equal to 2. Three reviewers (N.A.O., M.N.U., and M.L.B.) independently assessed the methodological quality of 21 studies, and the S5 File comprises the reviewers' evaluation sheets. After discussion, we excluded the publications all three reviewers rated as "poor" in methodological quality. A flawed framework would have introduced bias into the analysis, resulting in an inaccurate representation of our retrieved data items, notably the outcome. S4 File contains the excluded 7 articles that did not meet the threshold.

## Certainty assessment

Three reviewers (N.A.O., M.N.U., and M.L.B.) independently assessed the certainty of the outcome of the reviewed studies using the Grading of Recommendations Assessment,

**Table 2. Methodological quality appraisal tool.**

| Criterion | Details | Score |
|---|---|---|
| 1. Outcome measures | A. Valid/identifiable with well-defined objectives | 1 |
| | B. Identifiable, but with ambiguous terminologies and objectives | 0.5 |
| | C. Invalid/unreliable and poorly described | 0 |
| 2. Background or literature review | A. Detailed, with discernible differences from the reviewed studies | 1 |
| | B. Detailed, but no distinguishing feature from the reviewed studies is indicated | 0.5 |
| | C. Limited or non-existent | 0 |
| 3. Sample or dataset | A. Well-described data collecting process, with substantial data for validation | 1 |
| | B. Well-explained data collecting process, with inadequate data for validation | 0.5 |
| | C. Ineptly outlined data collecting process, with almost no, if any, data for validation | 0 |
| 4. Study design or methodology | A. Straightforward, comprehensible, and reproducible | 1 |
| | B. Fairly clear, and but not completely reproducible | 0.5 |
| | C. Ambiguous framework and an absence of reproducibility | 0 |
| 5. Conclusions | A. Completely substantiated by the findings | 1 |
| | B. Mostly substantiated, yet there are some undefined and partially reported findings | 0.5 |
| | C. Not supported by the results, with unverified presumptions | 0 |

Development and Evaluation (GRADE) defined four levels of evidence quality (see S1 Table) [21]. S6 File comprises the certainty assessment sheet for evaluating the 14 studies selected for this systematic review.

## Results

### Study selection

Fig 1 illustrates the complete PRISMA 2020 framework for the selection process following the eligibility criteria. We discovered 2,399 non-duplicate records while scanning the databases. Only 44 reports remained after the screening, and we were unable to retrieve 2 of them. We

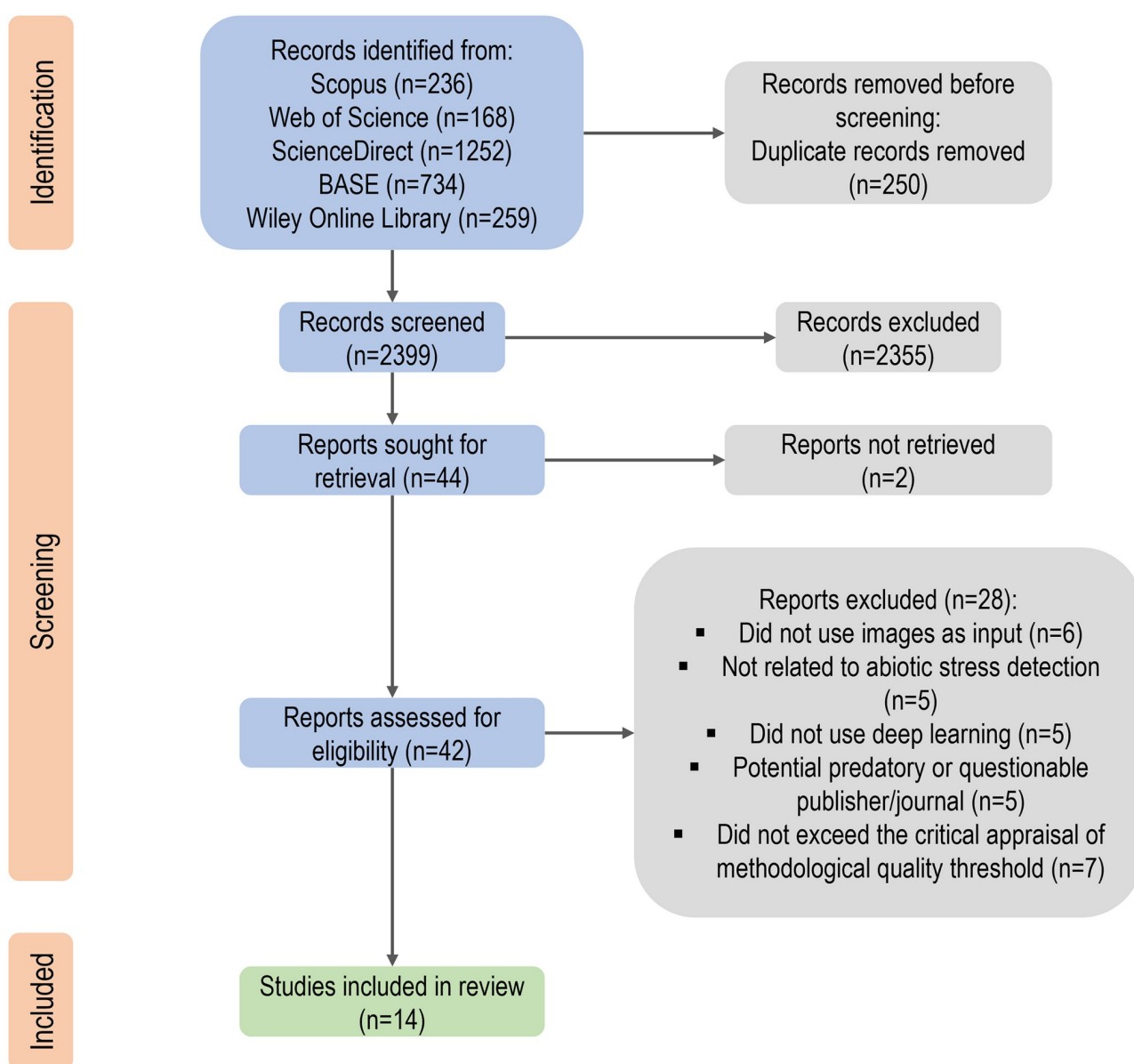

**Fig 1. PRISMA 2020 workflow of the systematic study.** The hierarchical figure depicts the complete number of entries evaluated at various phases of the systematic review, as well as the reasons for exclusion in the final step of screening.

eventually considered 14 papers [22–35] eligible and in-topic for the systematic review after reviewing the 42 reports. S4 File depicts the specific rationale behind the exclusion of the remaining reports.

## Study characteristics

In this section, we provide an analytical overview of all the reviewed articles, highlighting major aspects of the retrieved data items. Fig 2 illustrates the number of selected studies dispersed over the years. Prior to 2018, there appeared to be a lack of research in this subject, despite the effective use of deep learning and computer vision in biomedical [36] and other agricultural [37, 38] domains. Nevertheless, this sector is growing steadily because the majority of study is from 2022.

Regarding the crops chosen, Fig 3 displays the distribution among the studies. Rice (*Oryza sativa* L.) is one of the most researched crops for detecting abiotic stresses, which is fitting given that it is the most essential crop in the world, sustaining more than half of the planet's population [39]. While Sugarcane (*Saccharum officinarum* L.), Lettuce (*Lactuca sativa* L.), and Maize (*Zea mays* L.) have only been studied once each, Tomato (*Solanum lycopersicum* L.), Soybean (*Glycine max* L. Merr.), and Sugar beet (*Beta vulgaris* L.) have all undergone multiple investigations.

Fig 4 shows the distribution of the abiotic stressors discussed in the articles included in the systematic review. Thirteen of the fourteen research papers discussed potassium deficiency. The top three addressed abiotic stressors, as shown in Fig 4, are also the most important macronutrients for plant growth [40], collectively known as NPK.

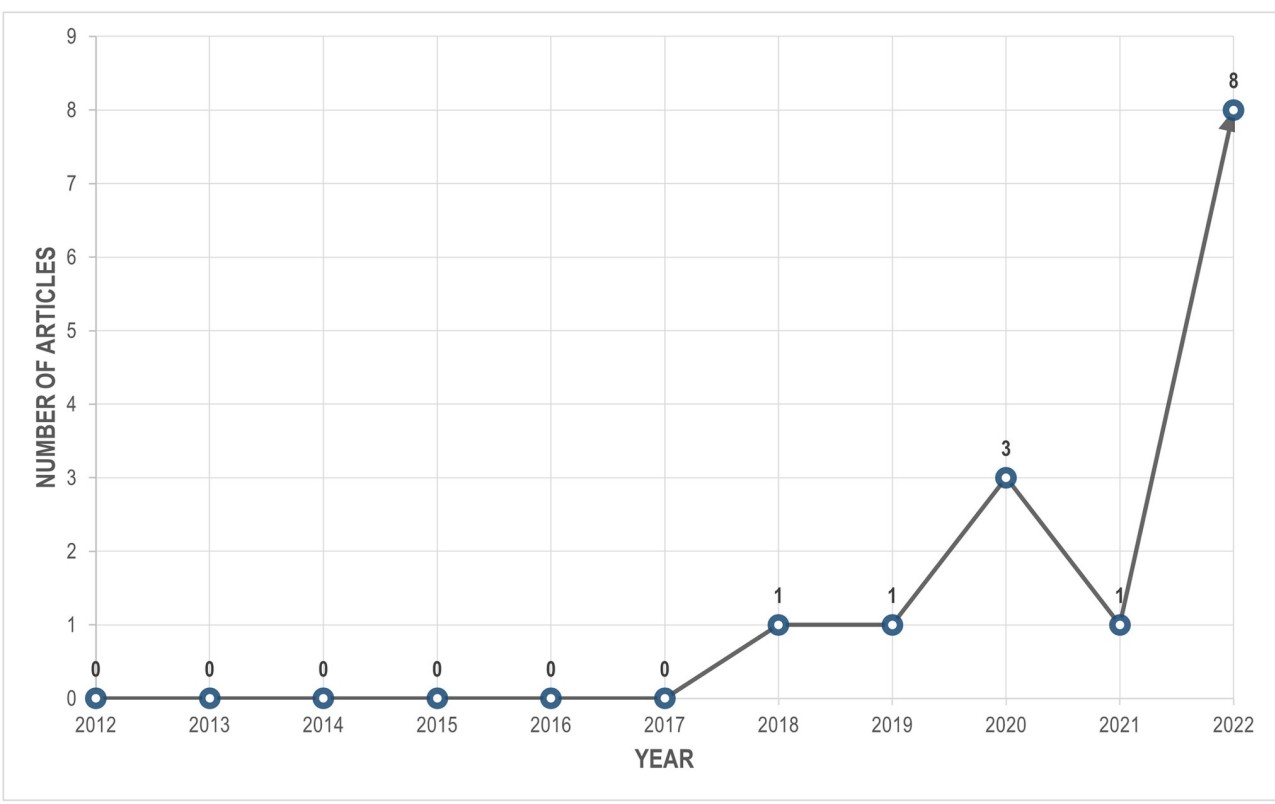

**Fig 2. Temporal distribution of research articles in the systematic review.** It should be noted that the calculated distribution of the research extends only to December 18, 2022.

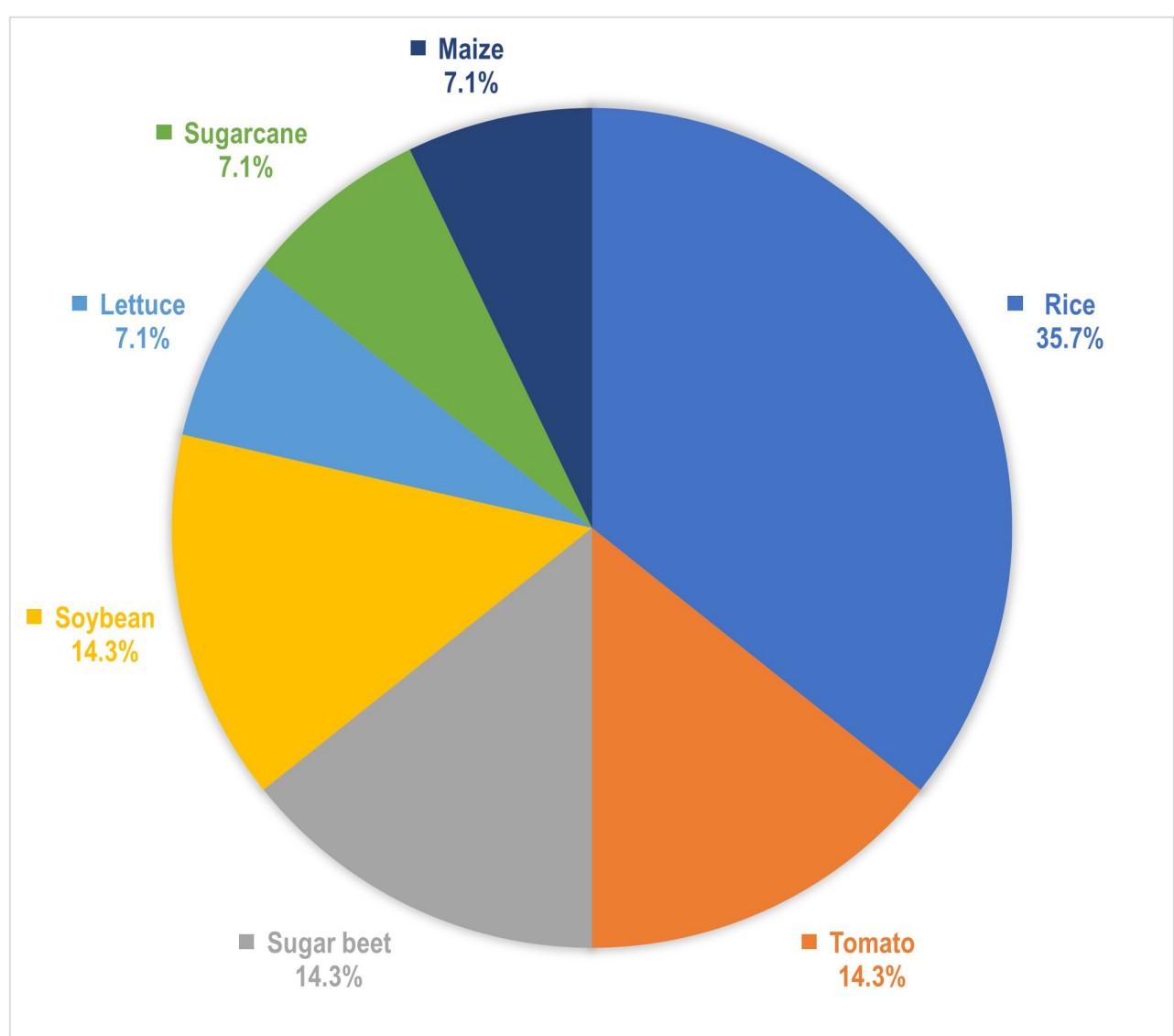

**Fig 3. Distribution of crops targeted by selected research articles.** Rice has drawn the most research interest (5 out of 14 studies), followed by Tomato, Soybean, and Sugar beet (2 out of 14 studies each).

Table 3 outlines the characteristics of the datasets featured in the included studies. The samples were generally collected in two ways. The first method used field-based sampling, and the stressors were not induced in the crops, but rather observed spontaneously. In the second procedure, stressors such as drought or a deficiency of vital nutrients were artificially induced in the targeted crops. Both strategies have a roughly similar number of studies, with 6 employing the initial method and 8 using the latter. The RGB color model, which has the primary colors red, green, and blue divided into three channels, was used in all of the articles under review. With the exception of [27], which employed thermal pictures for the cognition of drought stress in sugarcane, every single reviewed report employed digital images. As shown in Table 3, there is a definite indication of an equitable distribution of study for both open-access and closed-access datasets (7 studies each).

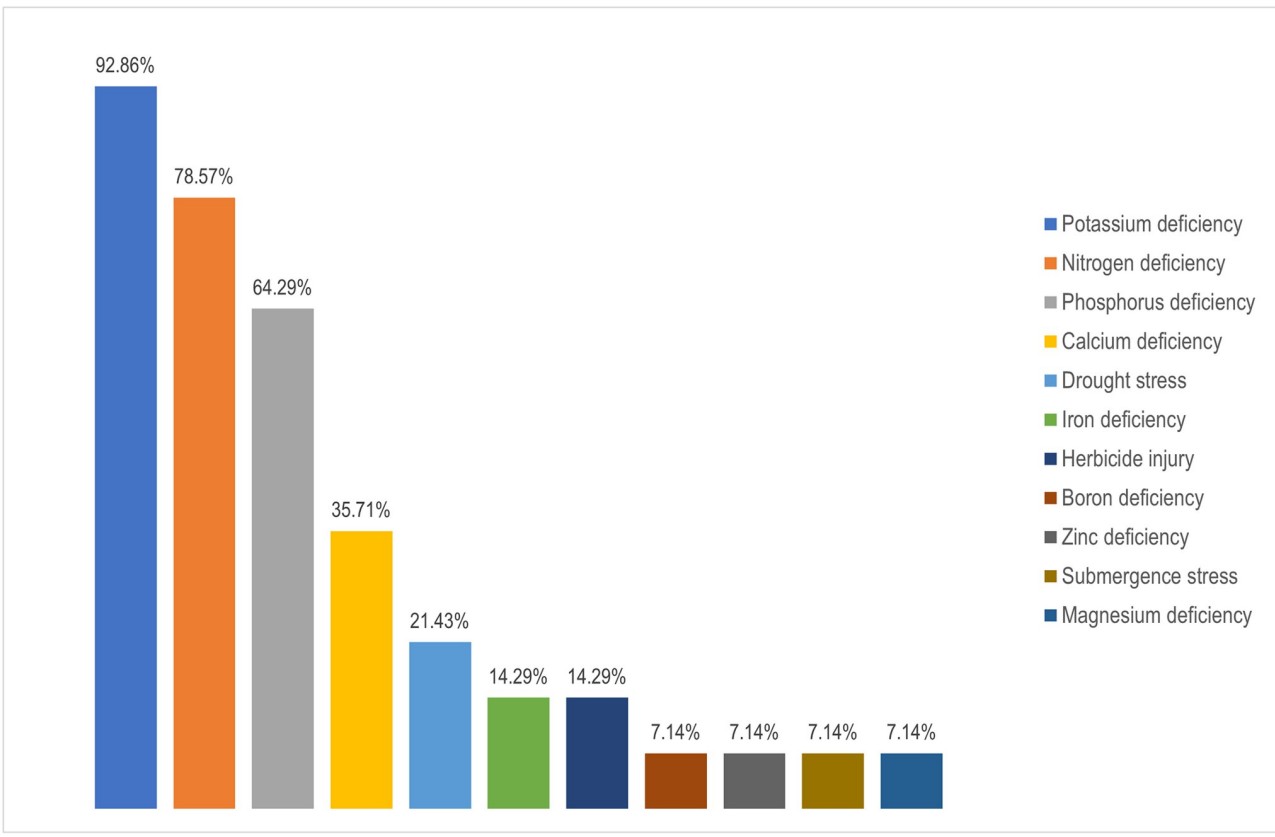

**Fig 4. Distribution of crop abiotic stressors targeted by selected research articles.** The graph displays the percentage of overall research addressing a specific abiotic stressor divided by the number of publications considered for the systematic review. It should be noted that several research targeted multiple abiotic stressors simultaneously.

The dispersion of the reviewed articles according to the deep learning frameworks that each study indicated to be most effective is illustrated in Fig 5. Ensemble learning integrates predictions from several models of neural networks to minimize the prediction variance and generalization error [41]. The biggest drawback of selecting just one top-performing model is that it may not always be the network that performs the best on unseen test data [42]. Hence, of the 14 studies considered, 4 used ensembling to combine various pre-trained DCNN models to offer the best and most efficient classifiers. A few research have made use of custom-built DCNN structures as well, as shown in Fig 5.

Fig 6 represents the various pre-trained networks and the instances that they were used in this systematic review. The predominantly implemented deep learning architectures in the selected studies were InceptionV3 [43], InceptionResNetV2 [44], and DenseNet201 [45] (3 out of 14 studies each). The reviewed articles also used VGG16 and VGG19 [46], Xception [47], and other DenseNet structures, as depicted in Fig 6.

Table 4 summarizes the integrated approach, data partition, and learning strategy adopted in the included research. All research employed supervised learning, except for [22], where the authors utilized a subset of the data for training the Autoencoder model unsupervised before combining it with supervised InceptionResNetV2. Table 4 indicated a heterogeneity in split ratio among studies. A dataset is divided into training, validation, and testing subsets to assist a model in optimizing its weights, avoiding overfitting and underfitting, and establishing robustness by assessing performance on unseen data, respectively [48]. Instead of a fully connected

**Table 3. An overview of the dataset features in the reviewed articles.**

| Ref. | Crop | Studied abiotic stressors | Sample size | Sampling condition | Image type | Color space | Data Accessibility |
|------|------|---------------------------|-------------|--------------------|------------|-------------|--------------------|
| [22] | Tomato | Calcium deficiency | 305 | Controlled | Digital | RGB | Not open |
| | | Potassium deficiency | 131 | | | | |
| | | Nitrogen deficiency | 135 | | | | |
| [23] | Soybean | Iron deficiency | 1834 | On-field | Digital | RGB | Not open |
| | | Potassium deficiency | 2182 | | | | |
| | | Herbicide injury | 1311 | | | | |
| [24] | Rice | Phosphorus deficiency | 333 | On-field | Digital | RGB | Open |
| | | Potassium deficiency | 383 | | | | |
| | | Nitrogen deficiency | 440 | | | | |
| [25] | Rice | Phosphorus deficiency | 333 | On-field | Digital | RGB | Open |
| | | Potassium deficiency | 383 | | | | |
| | | Nitrogen deficiency | 440 | | | | |
| [26] | Rice | Phosphorus deficiency | 500 | Controlled | Digital | RGB | Not open |
| | | Potassium deficiency | 500 | | | | |
| | | Nitrogen deficiency | 500 | | | | |
| | | Boron deficiency | 500 | | | | |
| | | Zinc deficiency | 500 | | | | |
| | | Iron deficiency | 500 | | | | |
| | | Herbicide injury | 500 | | | | |
| | | Drought stress | 500 | | | | |
| | | Submergence stress | 500 | | | | |
| [27] | Sugarcane | Drought stress | 1350 | Controlled | Thermal | RGB | Not open |
| [28] | Sugar beet | Nitrogen deficiency | 708 | Controlled | Digital | RGB | Open |
| | | Phosphorus deficiency | 808 | | | | |
| | | Potassium deficiency | 794 | | | | |
| | | Calcium deficiency | 893 | | | | |
| [29] | Rice | Phosphorus deficiency | 333 | On-field | Digital | RGB | Open |
| | | Potassium deficiency | 383 | | | | |
| | | Nitrogen deficiency | 440 | | | | |
| [30] | Tomato | Nitrogen deficiency | 103 | Controlled | Digital | RGB | Not open |
| | | Magnesium deficiency | 152 | | | | |
| | | Potassium deficiency | 223 | | | | |
| | | Calcium deficiency | 207 | | | | |
| [31] | Maize | Drought stress | 6320 | Controlled | Digital | RGB | Open |
| [32] | Rice | Phosphorus deficiency | 333 | On-field | Digital | RGB | Open |
| | | Potassium deficiency | 383 | | | | |
| | | Nitrogen deficiency | 440 | | | | |
| [33] | Soybean | Potassium deficiency | 1083 | On-field | Digital | RGB | Not open |
| [34] | Lettuce | Phosphorus deficiency | 550 | Controlled | Digital | RGB | Not open |
| | | Potassium deficiency | 1000 | | | | |
| | | Nitrogen deficiency | 850 | | | | |
| [35] | Sugar beet | Nitrogen deficiency | 708 | Controlled | Digital | RGB | Open |
| | | Phosphorus deficiency | 808 | | | | |
| | | Potassium deficiency | 794 | | | | |
| | | Calcium deficiency | 893 | | | | |

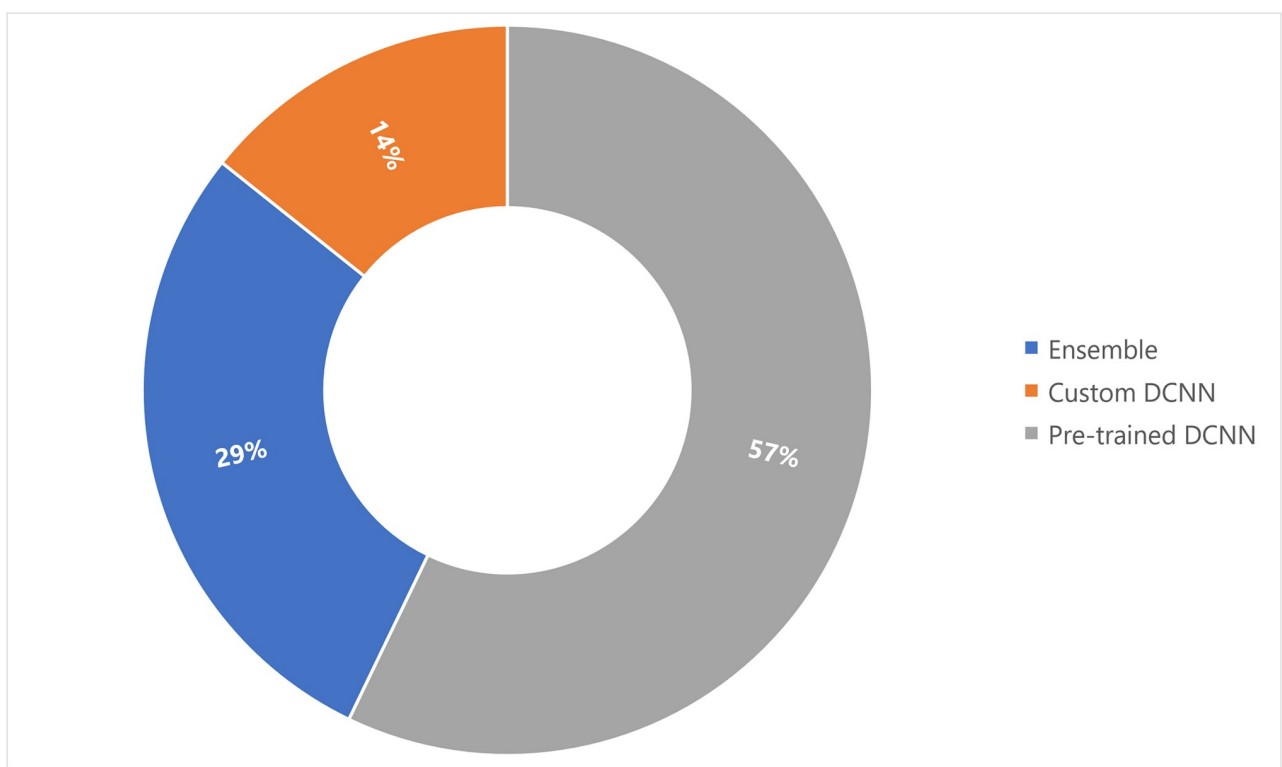

**Fig 5. Distribution of deep learning architectures employed by selected research articles.** The authors primarily employed the most common pre-trained networks accessible at present (8 out of 14 studies), with custom-designed networks being used the least (2 out of 14 studies).

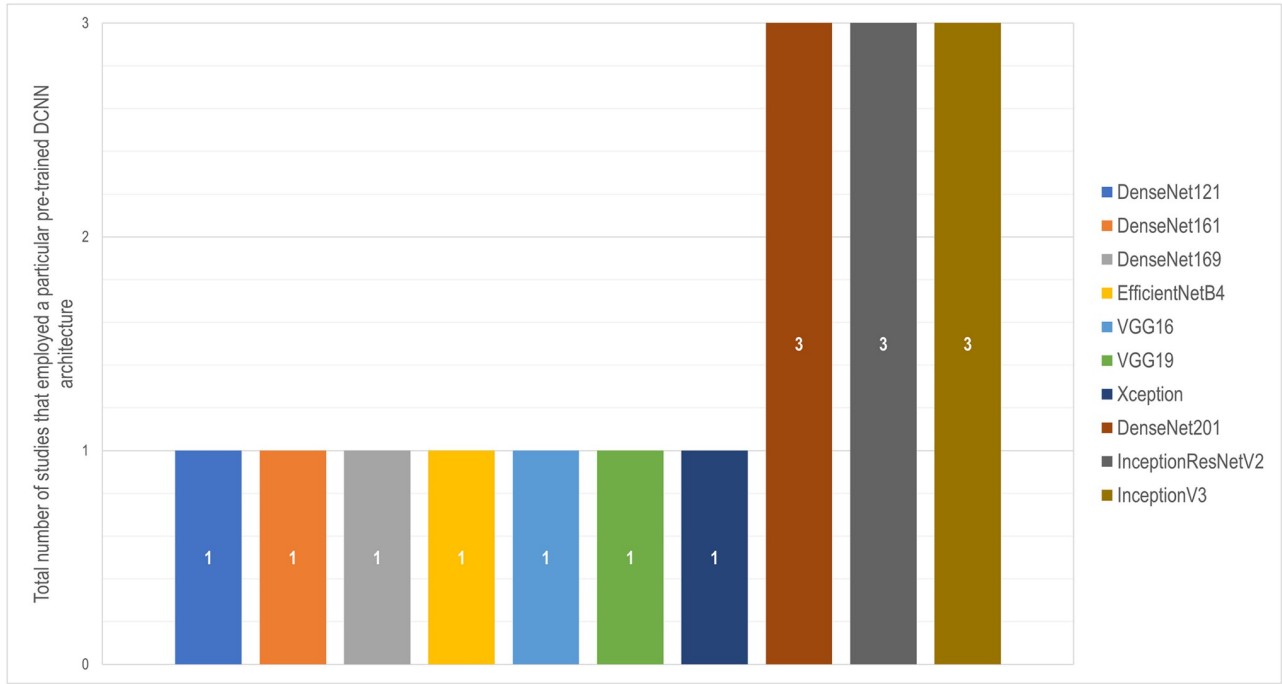

**Fig 6. Distribution of pre-trained DCNN architectures employed by selected research articles.** It should be emphasized that several studies employed multiple pre-trained DCNN architectures for ensembling.

**Table 4. An overview of the strategies implemented in the articles under review.**

| Ref. | Dataset split ratio | Best performing framework | Abbr. framework | Learning type |
|---|---|---|---|---|
| [22] | 0.80:0.20:0 | Ensemble (InceptionResNetV2 and Autoencoder) | EM1 | Semi-supervised |
| [23] | 0.70:0.20:0.10 | Custom DCNN | C1 | Supervised |
| [24] | 0.70:0.10:0.20 | VGG19 | PT1 | Supervised |
| [25] | 0.80:0.10:0.10 | Ensemble (InceptionV3 and Xception) | EM2 | Supervised |
| [26] | 0.60:0.04:0.36 | VGG16 | PT2 | Supervised |
| [27] | 0.70:0.30:0 | InceptionResNetV2 | PT3 | Supervised |
| [28] | 0.70:0.30:0 | DenseNet161 | PT4 | Supervised |
| [29] | 0.73:0.18:0.09 | Ensemble (InceptionResNetV2 and DenseNet201) | EM3 | Supervised |
| [30] | 0.72:0.18:0.10 | DenseNet121 | PT5 | Supervised |
| [31] | 0.53:0.13:0.34 | Custom DCNN with SVM | C2 | Supervised |
| [32] | 0.75:0.20:0.05 | Ensemble (DenseNet169, DenseNet201, and InceptionV3) | EM4 | Supervised |
| [33] | 0.85:0.05:0.10 | DenseNet201 | PT6 | Supervised |
| [34] | 0.60:0.20:0.20 | InceptionV3 | PT7 | Supervised |
| [35] | 0.80:0.10:0.10 | EfficientNetB4 | PT8 | Supervised |

layer, the authors employed Support Vector Machines (SVM) for classification in [31]. We have abbreviated the titles of the frameworks to render them simpler to recall in the following parts.

Table 5 discloses the highest precision and recall attained for each crop and its associated abiotic stressors. According to Table 5, Rice happens to be the most diversified, having been

**Table 5. An overview of the outcomes in the reviewed articles.**

| Crop | Studied abiotic stressors | Highest precision (in %) | Corresponding model | Highest recall (in %) | Corresponding model |
|---|---|---|---|---|---|
| Rice | Phosphorus deficiency | 100 | EM4 [32] | 100 | EM4 [32] |
| | Potassium deficiency | 100 | EM4 [32] | 95.24 | EM4 [32] |
| | Nitrogen deficiency | 95.46 | EM4 [32] | 100 | EM4 [32] |
| | Boron deficiency | 95.81 | PT2 [26] | 91.5 | PT2 [26] |
| | Zinc deficiency | 97.41 | PT2 [26] | 94 | PT2 [26] |
| | Iron deficiency | 96.46 | PT2 [26] | 95.5 | PT2 [26] |
| | Herbicide injury | 91.39 | PT2 [26] | 95.5 | PT2 [26] |
| | Drought stress | 96.06 | PT2 [26] | 97.5 | PT2 [26] |
| | Submergence stress | 94.03 | PT2 [26] | 94.5 | PT2 [26] |
| Tomato | Nitrogen deficiency | 83.16 | PT5 [30] | 83.61 | PT5 [30] |
| | Magnesium deficiency | 84.91 | PT5 [30] | 91.84 | PT5 [30] |
| | Potassium deficiency | 92 | PT5 [30] | 91.09 | PT5 [30] |
| | Calcium deficiency | 100 | PT5 [30] | 98 | PT5 [30] |
| Sugar beet | Nitrogen deficiency | 99.99 | PT8 [35] | 98.91 | PT8 [35] |
| | Phosphorus deficiency | 100 | PT8 [35] | 98.37 | PT8 [35] |
| | Potassium deficiency | 99.43 | PT8 [35] | 99.54 | PT4 [28] |
| | Calcium deficiency | 99.64 | PT8 [35] | 99.11 | PT8 [35] |
| Soybean | Iron deficiency | 97.99 | C1 [23] | 99.06 | C1 [23] |
| | Potassium deficiency | 100 | PT6 [33] | 100 | PT6 [33] |
| | Herbicide injury | 96.99 | C1 [23] | 98.02 | C1 [23] |
| Lettuce | Phosphorus deficiency | 94.34 | PT7 [34] | 90.91 | PT7 [34] |
| | Potassium deficiency | 97 | PT7 [34] | 97 | PT7 [34] |
| | Nitrogen deficiency | 96.49 | PT7 [34] | 97.06 | PT7 [34] |
| Sugarcane | Drought stress | 81.11 | PT3 [27] | 87.69 | PT3 [27] |
| Maize | Drought stress | 96.74 | C2 [31] | 89 | C2 [31] |

subjected to a multitude of distinct abiotic stressors. Every single one of the research achieved precision and recall scores of more than 80%, with several research studies reaching 100% for certain abiotic stressors. Among the studies included, the models struggled to recognize nitrogen deficiencies in Tomato, achieving the lowest pair of precision and recall scores. On the opposite end of the scale, the associated deep learning frameworks achieved 100% on both metrics for Rice phosphorus deficiency and Tomato potassium deficiency. However, concerning the latter research, it should be highlighted that the authors tested merely 5% of the dataset, which equates to approximately 16 samples for the corresponding class.

## Discussion

This section highlights the research gaps identified during our systematic review. It is worth noting that the research gaps we are addressing strictly fall within the scope of our research, which is geared towards image-based deep learning frameworks for crop abiotic stress assessment.

1. **Crops:** According to an FAO report, over 6000 plant species have been grown for food, with 200 species maintaining a considerable production level globally [49]. As seen in Fig 3, we came across only 7 of these crops throughout our systematic review. Moreover, the world's top five staple foods—Rice, Wheat (*Triticum aestivum* L.), Maize, Potato (*Solanum tuberosum* L.), and Cassava (*Manihot esculenta* Crantz)—provide the bulk of the world's dietary requirement for both nutrients and energy [50]. Although we found some studies on Rice and Maize, we encountered no publications on Wheat, Potato, or Cassava.

2. **Abiotic stressors:** According to Fig 4, the majority of the research targeted various micro and macronutrient deficiencies in crops. Some studies also investigated water-related stresses including drought and submergence. Unfortunately, no research on the cognition of early indicators of water stress or nutritional inadequacies were encountered. Other abiotic factors that have a detrimental effect on crop productivity include soil characteristics such as acidity, alkalinity, and salinity [51, 52], as well as variations in light intensity [53]. Furthermore, our planet is undergoing a significant climatic transition, with temperature as a key indicator. According to this systematic study, no literature addressed temperature-related stress that affects plant physiological mechanisms [54].

3. **Datasets:** In precision agriculture, the labor and expenditures necessary for image acquisition, categorization, and labeling, as well as physico-chemical evaluations of crops in some circumstances, end up making dataset preparation challenging [55]. Open access data reduces the difficulty of data preparation while also ensuring reproducibility and encouraging more individuals to take part. For instance, 4 out of 5 Rice-related research used the same dataset, which is freely available on Kaggle. In the case of Sugar beet, a research released the dataset alongside their article, which prompted another study on the same dataset by different authors. Notwithstanding the upsurge of research seen in Fig 2, additional open access datasets can draw researchers' interest in this subject matter. Furthermore, the datasets presented only contain RGB pictures, although alternative color spaces have achieved state-of-the-art scores in individual plant diseases [56] and on the benchmark PlantVillage dataset [57].

4. **Deep learning architectures:** Although certain DenseNet, VGG, and Inception-based architectures were employed, a multitude of DCNNs remained left out. For example, just one of the approximately 15 alternative topologies from the EfficientNet [58] and

EfficientNetV2 [59] networks have been utilized (see Fig 6). We found no publications that used MobileNet frameworks, which are considered to be computationally efficient and resource-friendly [60]. Moreover, ConvNeXt architectures [61] and Vision Transformers (ViT) [62], which hold state-of-the-art for the majority of benchmark datasets and are readily available through Python's Keras [63] and Pytorch [64] libraries, respectively, weren't used in any of the studies.

## Conclusion

Our research aimed to present a holistic view of crop abiotic stress cognition using cutting-edge deep learning techniques in computer vision. With this in mind, we conducted a systematic review of studies spanning the last 10 years using the PRISMA framework. We found 14 publications after conducting the systematic search, which we used to highlight current developments in the subject. Rice and potassium shortage represent the most researched crop and abiotic stress. The authors preferred DenseNet topologies, with Inception models following close behind. We extensively outlined research shortcomings that could potentially be resolved.

We want to be clear that even though we followed the procedure meticulously, it is entirely possible that we missed essential reports or increased the chance of error. However, we are adamant that none of these restrictions will affect the review's overall substance. This study can guide computer and plant science researchers, encouraging further work.

## Supporting information

**S1 File. PRISMA 2020 checklist.** This document includes the updated PRISMA checklist as well as the position or page number of each item reported.
(DOCX)

**S2 File. Complete search terms with additional parameters.** This file contains the whole search string for each database considered, as well as the extra criteria used to retrieve relevant entries.
(DOCX)

**S3 File. Data extraction form.** This file features a data collecting sheet that was used to extract relevant data items from the manuscripts.
(XLSX)

**S4 File. Criteria of the omitted studies and their justifications.** This file includes the specific reasoning for each study that was excluded after accessing the complete texts.
(DOCX)

**S5 File. Evaluation sheets of the studies.** This document comprises the reviewers' ratings of the studies we considered for methodological quality appraisal.
(XLSX)

**S6 File. Certainty assessment sheet of the studies.** This file includes the reviewers' assessments of the research outcomes or findings following the GRADE tool.
(XLSX)

**S1 Fig. The decision tree diagram for screening possible predatory or doubtful journals/ publishers.** Using this chart, we evaluated the publishers or journals depending on their expulsion from Scopus coverage for editorial malpractices, false and misleading information

regarding indexing and memberships on their websites, a lack of overall transparency, and a variety of other factors.
(TIF)

**S1 Table. The four levels of evidence used in the GRADE profile.** Using this table, we assessed the findings of the reviewed studies, namely the inter-class precision and recall, while taking into account GRADE's five identified categories: risk of bias, imprecision, inconsistency, indirectness, and publication bias.
(DOCX)

## Acknowledgments

We would like to express our sincere gratitude to Mahmud Hossain Al-Mamun for his thoughtful contributions and insights throughout the systematic research.

## Author Contributions

**Conceptualization:** Nabil Anan Orka, M. Nazim Uddin.

**Data curation:** Nabil Anan Orka, Fardeen Md. Toushique, M. Nazim Uddin.

**Formal analysis:** Fardeen Md. Toushique.

**Investigation:** Nabil Anan Orka, M. Nazim Uddin.

**Methodology:** Nabil Anan Orka.

**Project administration:** M. Latiful Bari.

**Resources:** Nabil Anan Orka, M. Nazim Uddin.

**Software:** Nabil Anan Orka, M. Nazim Uddin.

**Supervision:** M. Nazim Uddin, M. Latiful Bari.

**Validation:** M. Nazim Uddin, M. Latiful Bari.

**Visualization:** Nabil Anan Orka.

**Writing – original draft:** Nabil Anan Orka.

**Writing – review & editing:** Fardeen Md. Toushique, M. Nazim Uddin, M. Latiful Bari.

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
