## [Decision Letter · Decision Letter 0]

13 Jul 2023

PONE-D-23-09594Application of computer vision in assessing crop abiotic stress: A systematic reviewPLOS ONE

Dear Dr. Nabil Anan Orka

Thank you for submitting your manuscript to PLOS ONE. After careful consideration, we feel that it has merit but does not fully meet PLOS ONE’s publication criteria as it currently stands. Therefore, we invite you to submit a revised version of the manuscript that addresses the points raised during the review process. Please submit your revised manuscript by Aug 27 2023 11:59PM. If you will need more time than this to complete your revisions, please reply to this message or contact the journal office at plosone@plos.org. Please include the following items when submitting your revised manuscript:A rebuttal letter that responds to each point raised by the academic editor and reviewer(s). You should upload this letter as a separate file labeled 'Response to Reviewers'.A marked-up copy of your manuscript that highlights changes made to the original version. You should upload this as a separate file labeled 'Revised Manuscript with Track Changes'.An unmarked version of your revised paper without tracked changes. You should upload this as a separate file labeled 'Manuscript'.If applicable, we recommend that you deposit your laboratory protocols in protocols.io to enhance the reproducibility of your results. Protocols.io assigns your protocol its own identifier (DOI) so that it can be cited independently in the future. For instructions see: https://journals.plos.org/plosone/s/submission-guidelines#loc-laboratory-protocols. Additionally, PLOS ONE offers an option for publishing peer-reviewed Lab Protocol articles, which describe protocols hosted on protocols.io. Read more information on sharing protocols at https://plos.org/protocols?utm_medium=editorial-email&utm_source=authorletters&utm_campaign=protocols.

We look forward to receiving your revised manuscript.

Kind regards,

Wazir Muhammad

Academic Editor

PLOS ONE

Journal Requirements:

Reviewers' comments:

Reviewer's Responses to Questions

**Comments to the Author**

1. Is the manuscript technically sound, and do the data support the conclusions?

Reviewer #1: Yes

Reviewer #2: Yes

2. Has the statistical analysis been performed appropriately and rigorously? 

Reviewer #1: Yes

Reviewer #2: Yes

3. Have the authors made all data underlying the findings in their manuscript fully available?

Reviewer #1: Yes

Reviewer #2: Yes

4. Is the manuscript presented in an intelligible fashion and written in standard English?

Reviewer #1: Yes

Reviewer #2: Yes

5. Review Comments to the Author

Reviewer #1: The review article entitled Application of computer vision in assessing crop abiotic stress: A systematic review by Nabil Anan Orka et al is very well written.

The title of the manuscript is quite good and can attract readers attention.

I would recommend acceptance of the manuscript.

Reviewer #2: Following an extensive screening and analysis of papers published in the past decade, a comprehensive total of 2399 records from five search sources underwent screening for this study. Ultimately, a final selection of 14 publications was yielded. The total publication records offer valuable insights into 11 abiotic stressors that impact seven distinct crops. It is worth noting that a notable majority of the selected publications (8 out of 14) originate from 2022, with no publications spanning the years 2012 to 2017.

There are some contents that are not very clear to me, please clarify them in writings or figures.

1) In Figure 1, there are certain factors that seem unclear, such as "Report not retrieved (n=2)" and "Did not use deep learning (n=5)". Could you please provide a more detailed explanation regarding why these publications were excluded from the list after the screening process?

2) Regarding Figure 2, 3, and 4, it appears that only the final yield of 14 publications was used to generate Figures 2 and 3. Are the 2399 total records distributed similarly across different years and plant species? Conversely, it seems that the total records were used to create Figure 4, aiming to identify the crop abiotic stressors. Therefore, I suggest using the total records to create Figures 2 and 3 as well. Similarly, the yield of the 14 selected publications could be used to create Figure 4.

3) Grammer error: “Several research” row 249, can be change to “several research studies”

4) Some references are in different styles. For example: reference 7 is "nature" while reference 16 is "Nature". Please correct this type of error accordingly.

6. PLOS authors have the option to publish the peer review history of their article (what does this mean?). If published, this will include your full peer review and any attached files.

Reviewer #1: **Yes: **Dr. Tariq Aziz

Reviewer #2: No

---

## [Author Response · Author response to Decision Letter 0]

17 Jul 2023

Reviewer #1: The review article entitled Application of computer vision in assessing crop abiotic stress: A systematic review by Nabil Anan Orka et al is very well written. The title of the manuscript is quite good and can attract readers attention. I would recommend acceptance of the manuscript.

 We would like to convey our sincerest gratitude for the kind words. It will be a privilege to represent a prestigious journal such as PLOS ONE.

Reviewer #2: Following an extensive screening and analysis of papers published in the past decade, a comprehensive total of 2399 records from five search sources underwent screening for this study. Ultimately, a final selection of 14 publications was yielded. The total publication records offer valuable insights into 11 abiotic stressors that impact seven distinct crops. It is worth noting that a notable majority of the selected publications (8 out of 14) originate from 2022, with no publications spanning the years 2012 to 2017. There are some contents that are not very clear to me, please clarify them in writings or figures. 

1) In Figure 1, there are certain factors that seem unclear, such as "Report not retrieved (n=2)" and "Did not use deep learning (n=5)". Could you please provide a more detailed explanation regarding why these publications were excluded from the list after the screening process? 

 We adhered to the PRISMA 2020 guidelines to conduct the systematic review. We chose the research articles for the systematic review (14 in total) by meticulously sticking to the rules and outlining the selection approach in detail. It should be emphasized that we did not omit studies from 2012 to 2017 on purpose, but rather that the deep learning boom in the stress identification space began late. When reviewers are unable to access articles from their individual online addresses, the issue must be indicated in the selection method/workflow figure, according to the standards. As a result, we indicated this using "Reports not retrieved," followed by the number of articles that could not be acquired, using the structure recommended by the PRISMA rules. In addition, we mentioned the number of inaccessible articles on lines 177-178. Our systematic review is exclusively geared towards image-based deep learning frameworks for crop abiotic stress assessment. As such, on lines 24-37, we explained why we chose deep learning, focusing on its advantages over traditional machine learning methods. Deep learning can recognize complicated patterns and extract significant information through minimal engineering, and in certain cases, without the need for human supervision. Moreover, we pointed out that the use of deep learning in the eligibility criteria section, namely the inclusion criterion on lines 54-56. Hence, we excluded papers that did not utilize deep learning. Furthermore, we provided a supporting information file called S4 that covers the reasoning behind each manuscript that was rejected during the second screening step. 

2) Regarding Figure 2, 3, and 4, it appears that only the final yield of 14 publications was used to generate Figures 2 and 3. Are the 2399 total records distributed similarly across different years and plant species? Conversely, it seems that the total records were used to create Figure 4, aiming to identify the crop abiotic stressors. Therefore, I suggest using the total records to create Figures 2 and 3 as well. Similarly, the yield of the 14 selected publications could be used to create Figure 4. 

 First and foremost, please accept our sincere apologizes for any misunderstandings or ambiguity. Figures 2, 3, and 4 illustrate the yield of the 14 papers considered for the systematic review. Unfortunately, our phrasing accentuated the vagueness. As a result, we modified the caption of figure 4. We evaluated 2399 papers in total, resulting in 14 publications for the final systematic review. All of the figures and tables in the results section are founded on the selected 14 articles, not the entirety of 2399 articles. The guidelines of PRISMA's screening system are composed in such a way that we cannot depict the distribution of papers on crops and abiotic stresses without reading the manuscripts. To avoid biases, accessing the main content of the article is prohibited at the first round of screening. We would thus be violating the protocol if we presented the entire set of 2399 articles in the results section. Again, we apologize for any confusion.

3) Grammar error: “Several research” row 249, can be change to “several research studies” 

 We sincerely apologize for overlooking the grammatical problem in our initial submission. We rectified it right away. Thank you for your valuable suggestion.

4) Some references are in different styles. For example: reference 7 is "nature" while reference 16 is "Nature". Please correct this type of error accordingly.

 We apologize for overlooking the referencing problems. Thank you very much for bringing these items to our attention. Since then, we have followed the requirements on the PLOS ONE website and corrected them. We also changed the names of the journals to match those listed in the National Center for Biotechnology Information (NCBI) databases.

---

## [Decision Letter · Decision Letter 1]

8 Aug 2023

Application of computer vision in assessing crop abiotic stress: A systematic review

PONE-D-23-09594R1

Dear Dr. Orka,

We’re pleased to inform you that your manuscript has been judged scientifically suitable for publication and will be formally accepted for publication once it meets all outstanding technical requirements.

Kind regards,

Sathishkumar V E

Academic Editor

PLOS ONE

Additional Editor Comments (optional):

Reviewers' comments:

Reviewer's Responses to Questions

**Comments to the Author**

1. If the authors have adequately addressed your comments raised in a previous round of review and you feel that this manuscript is now acceptable for publication, you may indicate that here to bypass the “Comments to the Author” section, enter your conflict of interest statement in the “Confidential to Editor” section, and submit your "Accept" recommendation.

Reviewer #2: All comments have been addressed

2. Is the manuscript technically sound, and do the data support the conclusions?

Reviewer #2: Yes

3. Has the statistical analysis been performed appropriately and rigorously? 

Reviewer #2: Yes

4. Have the authors made all data underlying the findings in their manuscript fully available?

Reviewer #2: Yes

5. Is the manuscript presented in an intelligible fashion and written in standard English?

Reviewer #2: Yes

6. Review Comments to the Author

Reviewer #2: I am of the opinion that the authors have addressed all the comments I previously provided. Therefore, I recommend accepting this manuscript.

7. PLOS authors have the option to publish the peer review history of their article (what does this mean?). If published, this will include your full peer review and any attached files.

Reviewer #2: **Yes: **Ran Tian, PhD

---

## [Editor Report · Acceptance letter]

14 Aug 2023

PONE-D-23-09594R1 

Application of computer vision in assessing crop abiotic stress: A systematic review 

Dear Dr. Orka:

I'm pleased to inform you that your manuscript has been deemed suitable for publication in PLOS ONE. Congratulations! Your manuscript is now with our production department. 

Kind regards, 

on behalf of

Dr. Sathishkumar V E 

Academic Editor

PLOS ONE